# An Improved Network Time Protocol for Industrial Internet of Things [note 1]

**DOI:** 10.3390/s22135021

**Published:** 2022-07-03

**Authors:** Ting-Chao Hou, Lin-Hung Liu, Yan-Kai Lan, Yi-Ting Chen, Yuan-Sun Chu

**Affiliations:** Department of Electrical Engineering, Advanced Institute of Manufacturing with High-Tech Innovations, National Chung Cheng University, Chia-Yi 621301, Taiwan; ieetch@ccu.edu.tw (T.-C.H.); lovesleepbaby520@gmail.com (L.-H.L.); ggyy870224@gmail.com (Y.-K.L.); nihonn714@gmail.com (Y.-T.C.)

**Keywords:** NTP, time synchronization, clock skew, RTT, Internet of Things

## Abstract

In the industrial Internet of Things, the network time protocol (NTP) can be used for time synchronization, allowing machines to run in sync so that machines can take critical actions within 1 ms. However, the commonly used NTP mechanism does not take into account that the network packet travel time over a link is time-varying, which causes the NTP to make incorrect synchronization decisions. Therefore, this paper proposed a low-cost modification to NTP with clock skew compensation and adaptive clock adjustment, so that the clock difference between the NTP client and NTP server can be controlled within 1 ms in the wired network environment. The adaptive clock adjustment skips the clock offset calculation when the NTP packet run trip time (RTT) exceeds a certain threshold. The clock skew compensation addresses the inherent issue that different clocks (or oscillators) naturally drift away from each other. Both adaptive clock adjustment and clock skew compensation are environment dependent and device dependent. The measurement result in our experimental environment shows that the when the RTT threshold is set at 1.7 ms, the best synchronization accuracy is achieved.

## 1. Introduction

The industrial Internet of Things (IIoT) is the core of the entire Industry 4.0 development. It is a system formed by connecting various sensors, computers, and other industrial equipment through a network, and by sending messages between machines. IIoT allows for real-time monitoring of machine operations and can help with reducing human factors that affect production quality. In IIoT, time is a key element. All devices or machines in the network need to have a common time reference so that they can operate consistently and take time-critical actions within a precise time scale. At this stage, the synchronization accuracy requirement of the industrial Internet of Things is 1 ms [1].

NTP is a network protocol designed by Dr. David L. Mills of the University of Delaware, which utilizes packet transmission for time synchronization. NTP is applied to the Internet of Things, which allows different machines to synchronize to the same server, thereby achieving synchronization between machines. It usually achieves an accuracy of 1 millisecond in an ideal local area network environment, but can cause errors of tens of milliseconds (or more) when the network is congested. In order to solve this problem, S. Borenius et al. [2] used the 5G Core to allow the 4G network to have some 5G functions, and used the 4G network for NTP time synchronization. They proposed a new algorithm to improve the accuracy of NTP. Although NTP lacks a high accuracy, it is still widely used in IIoT because of cost considerations and ease of installation. Rahul Gore et al. [3] proposed a software-based clock synchronization method, called CoSiWiNeT, based on the random sample consensus (RANSAC) algorithm that used an iterative technique to estimate the correct offset from the observed noisy data. P. Ferrari et al. [1] estimated the network traffic on the NTP in the industrial Internet of Things and evaluated whether NTP is suitable for use in the industrial Internet of Things. Their commonly used devices compared NTP and SNTP (simple network time protocol), and concluded that in the current IIoT environment, the most suitable time synchronization scheme is NTP. In this paper, we proposed a low-cost modification to NTP with clock skew compensation and adaptive clock adjustment by skipping the clock offset calculation when the NTP packet run trip time (RTT) exceeded a certain threshold, so that the NTP synchronization time could be controlled within 1 ms in the wired network environment.

The rest of this paper is organized as follows. Section 2 first reviews NTP principles and then describes clock skew measurement and adaptive clock adjustment. Section 3 explains that the proposed adaptive clock adjustment is safe and provides proof. Section 4 illustrates the effectiveness of the proposed NTP improvement in synchronizing two clients. Section 5 presents the conclusions.

## 2. Proposed Methodology to Improve NTP

### 2.1. NTP Princinple

The basic synchronization principle of NTP [4] is as follows. As shown in Figure 1, an NTP client sends a request packet to the NTP server and the server replies with a response packet that contains two timestamps: one is the time it receives the request packet ($t_{2}^{\prime }$) and the other is the time it sends out the response packet ($t_{3}^{\prime }$). For the client, it knows the time that the request packet is sent ($t_{1}$) and the time that the response packet is received ($t_{4}$). After the client receives the response packet, it has all four timestamps and can then calculate its clock difference from the server. Time stamps $t_{2}^{\prime }$ and $t_{3}^{\prime }$ are expressed with a prime because they are based on the server clock, while $t_{1}$ and $t_{4}\;$ are based on the client clock. The difference between the server clock and the client clock is called the clock offset (or offset, in short), *θ*. This is what the NTP intends to find [5,6]. Let $\delta_{1}$ be the one-way delay (including packet transmission time and propagation delay) from the client to the server, and $\delta_{2}$ be the one-way delay from the server to the client. Then, we have
(1)$$t_{2}=t_{2}^{\prime }-\theta =t_{1}+\delta_{1}$$
(2)$$t_{4}=t_{3}+\delta_{2}=t_{3}^{\prime }-\theta +\delta_{2}$$
where $t_{2}\;$ and $\;t_{3}$ are what $t_{2}^{\prime }$ and $t_{3}^{\prime }$ would be if they are based on the client’s clock. Assuming symmetric one-way delays, i.e., $\delta_{1}=\delta_{2\;}=\delta ,\;$ (1) and (2) become
(3)$$t_{2}^{\prime }-\theta =t_{1}+\delta$$
(4)$$t_{4}=t_{3}^{\prime }-\theta +\delta$$

From (3) and (4), the clock offset θ can be derived as follows:(5)$$\theta =\frac{\left(t_{2}^{\prime }-t_{1})+(t_{3}^{\prime }-t_{4}\right)}{2}$$
and the round-trip delay time $\;\delta_{1}+\delta_{2}$ (not including the processing time, $t_{3}^{\prime }-t_{2}^{\prime }$, at the server) is as follows:(6)$$\delta_{1}+\delta_{2}=\left(t_{4}-t_{1}\right)-\left(t_{3}^{\prime }-t_{2}^{\prime }\right)$$

It is important to note that an assumption of symmetric one-way delays is made when calculating the offset *θ* in (5).

Equations (5) and (6) do not shed any light on whether or how the clock offset and the round-trip delay are related. However, they are actually correlated in some way. If the round-trip delay time increases, the probability of offset errors also increases, as shown in Figure 2, from the study by David Mills in RFC 1128 [7]. This could the reason that a larger round-trip delay may mean less symmetric one-way delay, thus Equation (5) is less accurate. When the NTP client derives the offset and adds it to the current local time, time synchronization with the server is completed [8]. The round-trip delay time calculation in Equation (6) indicate the time required to transmit the request packet and receive the response packet from the server. Its value also serves as an indicator of whether the corresponding offset calculation is relatively accurate or not.

In networking, a server is a machine that provides services. A client is the machine that seeks services from the server. An NTP server is therefore a server that runs the NTP code to provide time synchronization services. In a large-scale network, the NTP server architecture is hierarchical, with the root NTP server at the top and clients at the bottom. In between the root NTP server and client are multiple levels of intermediate NTP servers. We considered a reference architecture in our NTP study, as shown in Figure 3. A regional time server was set up in the local factory network. There were multiple machine tools in the factory. These machine tools needed to be synchronized within the 1 ms requirement and hence served the role of NTP clients. The regional time server was the one just above the client machine tools, and the external server was one level above the regional time server.

### 2.2. Clock Skew Measurement

In a network such as the Internet, the clock skew describes the difference in frequency (first derivative of offset with time) of different clocks within the network [9]. Clock skew refers to the difference in the time it takes for the clock signal to reach each digital circuit, and each computer’s clock signal arrives at its digital circuit at different times, so each computer will have a different clock skew. The purpose of measuring the clock skew in this paper is to calculate the difference between the client and the regional time server per second, and to then add the measured clock skew to the formula to be adjusted. In (7), clock_drift = clock_skew × elapsed_time, where elapsed_time is the time difference between the current time and the previous synchronization time.
local_time ≤ local_time + offset + clock_drift(clock_skew × elapsed_time)(7)

Figure 4 shows the clock skew measured by the same client computer relative to the regional time server. The client computer sends an NTP sync packet to the regional time server every 16 s for a total duration of 1600 s (that is, 100 packets are sent), and this process is repeated four times. The x-axis in Figure 4 is the index corresponding to the 100 packets during each measurement, and the y-axis is the relative time between the client machine and the regional time server.

The slopes of the four measurement results are −2.65 μs/s, −2.62 μs/s, and −2.57 μs/s, −2.68 μs/s, and the average of the four measurements is −2.63 μs/s [10]. If the client sets NTP to synchronize every 16 s, −2.63 μs/s multiplied by 16, is approximately equal to −42 μs, and this −42 μs is the clock_drift in Equation (7).

However, from Figure 5, it can be seen that at the fourth clock skew measurement, the packet encounters a small network delay. The slope from point *x* = 3 to point *x* = 4 in the figure becomes steeper to −5.75 μs/s, while the slope from point *x* = 4 to point *x* = 5 becomes smoother and positive at 0.44 μs/s. It can be seen from this that when there is a bigger network delay jitter, the synchronization result will be affected. This motivated us to propose a mechanism that nullifies certain NTP offset derivations and clock synchronization when the corresponding run trip delay times are higher than normal, in order to achieve more accurate synchronization.

### 2.3. Adaptive Clock Adjustment

Typically, an NTP client sends a synchronization packet to the local time server every 16 (or 32 or 64) s. Figure 6 and Figure 7 show the offset and round-trip delay time (or round-trip time; RTT) measurement results, respectively, from the first 50 synchronization packet exchanges at the 16 s interval in this paper [11]. It can be observed that the offset varies with the network round-trip time [12,13]. 

We conducted two NTP experiments (A and B), both of which always adjusted the local clock based on the derived offset from the NTP timing packet exchange in each 16 s interval, and compared them with the third experiment (C), which skipped the clock adjustment if the RTT (round-trip time) of the NTP timing packet exchange was above a certain threshold. The threshold was set at 1.5 ms tentatively. Experiment A was carried out from 12 noon to 1 p.m., while experiment B was carried from 1 p.m. to 2 p.m. Experiment C was carried out from 2 p.m. to 3 p.m. Figure 8 [14] is the flow chart of adaptive clock adjustment. It can be seen from Table 1 that the average offset in experiment C was reduced to within 0.1 ms from above 0.2 ms in A and B, and the maximum offset was also reduced to within 1 ms from above 3 ms in A and B, reaching the goal of clock accuracy for IIoT. Therefore, the mechanism of adaptive clock adjustment can effectively improve the synchronization accuracy.

### 2.4. Determining RTT Threshold

In the previous experiments, we chose the RTT threshold (or null threshold) for adaptive clock adjustment to be 1.5 ms. In this section, we study what RTT threshold is the most appropriate. The following experiments are all NTP packet exchanges with the local time server in 16 s, and the number of measurements is 5400 times a day. The null percentage in Table 2 represents the proportion in the 5400 times that timestamp data sets and offset derivations were nullified. It can be seen from Table 2 that the average offsets using the adaptive clock adjustment mechanism were at least one order of magnitude lower than that without the adaptive clock adjustment mechanism. From the maximum offset value, it can be seen that the adaptive clock adjustment mechanism nullified extraordinary offset values due to the large RTT variation, and therefore avoided making an incorrect synchronization move. Regarding the choice of an appropriate threshold value, we observed that the null percentages were all within 10% for threshold values from 1.5 ms to 1.85 ms. However, when the threshold was set to 1.85 ms, the maximum offset value exceeded the goal of 1 ms. We conclude that the threshold value in our environment should be lower than 1.85 ms. We chose to use 1.7 ms as the threshold value. This was also when the lowest average offset was obtained.

Note that network delays and jitters are mostly uncertain and may arise due to a number of factors, including traffic patterns, interference, buffer overload, overhearing, re-transmission, etc. Our proposed NTP improvement is therefore environment dependent (for dealing with network delay, jitter, etc.) and device dependent (for dealing with clock skew). The parameter values (ex., RTT threshold 1.7 ms in our test environment) required prior measurements of the RTT variations and clock skews based on the types of devices (NTP servers and clients) or the network environment in use.

## 3. Impact of Nullified Offset

When consecutive-derived offsets are nullified, the client clock will not be adjusted to the server clock for a period of time. In this section, we explain that the proposed adaptive clock adjustment is safe, and prove that the number of consecutive nulls is within a very safe range with a high probability, which will not cause the client clock to deviate from the server clock by more than 1 ms due to accumulated clock skews. Table 3 shows the statistics of offset, RTT, and null percentage from the one-day measurement with both the 16 s and 32 s intervals. The RTT threshold is set to 1.7 ms. We used these data to find a fitted probability distribution function and adopted a reasonable mathematical model to prove our claim. Describing measurement data with mathematical models makes it easier to further investigate the problem at hand without extensive measurements. In our case, we found a reasonably-fit probability function, the exponential function, to describe the measured inter-null intervals. With it, the probability of *n* (a large number greater 10) consecutive nulls happening will be a very rare event.

### 3.1. Expected Value and λ

We first calculated the expected value of the inter-null intervals. For example, Equation (8) is the expected value E(X) calculation for the inter-null interval with the 16 s interval. There were a total of 180 nulls in the data set. We labelled time zero as a reference null time. Therefore, there were also 180 inter-nulls (*x*), expressed in number of 16 s intervals. Among them, there were seventeen occurrences of one-interval inter-nulls, seven occurrences of two-interval inter-null, seven occurrences of three-interval inter-null, etc. The largest inter-null was 279, which had only one occurrence. In this particular measurement, 30.128 is the numerical value of the expected value. The null occurrence rate λ is the inverse of the expected value E(X)**.**
(8)$$E\left(X\right)=1\times \frac{17}{180}+2\times \frac{7}{180}+3\times \frac{7}{180}+\cdots +279\times \frac{1}{180\;}\;\approx 30.128\;$$
(9)$$\lambda =\frac{1}{E\left(X\right)\;}\approx 0.0332\;$$

### 3.2. Fitting of 16 s Inter-Null Distribution with Exponential Distribution

We first tried the fitting of the 16 s-based inter-null distribution with the exponential distribution. The formula of the cumulative distribution function (CDF) of a (shifted) exponential [15,16] distribution is
(10)$$\;F\left(x;\lambda \right)=\{\begin{array}{c} 1-e^{-\lambda \left(x-1\right)},\;\;x\geq 1\\ \;\;\;0,\;\;x<1 \end{array}$$

It can be seen from Figure 9 that the measured inter-null distribution approached an exponential distribution, but there were still gaps and the largest gap was 0.161 at inter-null = 18. We next looked for a better fit with an exponential distribution of the different expected value.

It can be seen from Figure 10 that the exponential distribution with an expected value of 23 was a better fit for the 16 s-based inter-null distribution. The maximum gap in CDF was reduced to 0.0683.

We also used the Q–Q plot to observe the fitting with the two exponential distributions. By comparing Figure 11 and Figure 12 [17], it can be clearly seen that the exponential distribution with an expected value of 23 was a better fit.

Based on the 16 s NTP packet exchange interval and the RTT threshold for offset nullification at 1.7 ms, we showed that the inter-null distribution approached an exponential distribution with an expected value equal to 23. Assuming that the clock drift between the client clock and the server clock was 40 μs/s over a 16 s interval, it would take 25 consecutive offset nullifications to cause the server clock and client clock to be off by 1 ms. The probability of this occurrence is low and can be calculated as follows. Based on the exponential distribution approximation, the probability of inter-null interval being 1 is 1/23 = 0.043. The probability that 25 consecutive nulls occurs is 0.043^25^ = 6.87 × 10 ^−35^. If this calculation is too conservative, we can also use the measurement data for the calculations. The measured probability of the inter-null interval being 1 is 17/180 = 0.094. The probability that 25 consecutive nulls occur is 2.13 × 10^−26^ (0.094^25^). It can be safely concluded that the offset nullification in the proposed adaptive clock adjustment is unlikely to cause the client clock and the server clock to be off by more than 1 ms because of clock skew.

The reason we did not use the exponential distribution with the expected value of 30.128 is that the measured inter-null distribution was close to but not identical to the exponential distribution. In order to fit the exponential distribution, a better-fitted (based on the overall distribution) exponential distribution, other than based on just one attribute (the expected value) of the distribution, needs to be selected.

### 3.3. Fitting of 32 s Inter-Null Distribution with Exponential Distribution

For the 32 s inter-null distribution, the measured expected value of the inter-null distribution is 24.897 using a method similar to Equation (8) and its λ is 0.0402. It can be seen from Figure 13 that the 32 s inter-null distribution is also close to the exponential distribution, but there are still some gaps between the two distributions, with the maximum gap being at 0.1334. We then tried different expected values for the exponential distribution and identified the expected value of 19.1 to be a good one. 

From Figure 14, it can be seen that the 32 s inter-null distribution was better fitted to an exponential distribution with an expected value equal to 19.1. The maximum gap is reduced to 0.0462, which is roughly three times smaller than when the expected value is 24.897.

Based on the 32 s NTP packet exchange interval and the RTT threshold for offset nullification at 1.7 ms, we show that the inter-null distribution approaches an exponential distribution with an expected value equal to 19.1. Assuming that the clock drift between the client clock and the server clock is 80 μs/s over a 32-s interval, it would take 12.5 consecutive offset nullifications to cause the server clock and client clock to be off by 1 ms. The probability of this occurrence can be calculated as follows. Based on the exponential distribution approximation, the probability of inter-null interval being one is 1/19.1 = 0.0524. The probability that 13 (the ceiling number of 12.5) consecutive nulls occur is 0.0524^13^ = 2.25 × 10^−17^. We can also use the measurement data for the calculation. The measured probability of the inter-null interval being 1 is 0.067. The probability that 13 consecutive nulls occur is 5.48 × 10^−15^ (0.067^16^). It can be safely concluded that the offset nullification in the proposed adaptive clock adjustment is unlikely to cause the client clock and the server clock to be off by more than 1 ms because of clock skew.

### 3.4. Comparison with Other Distributions

Visually, the measured inter-null distribution is close to the exponential distribution. In this section, we show that the exponential distribution is indeed more fitted than the other two commonly used distributions—the normal distribution and the Poisson distribution. We use the data set with the 16 s interval and the expected value is 30.128. Regarding the comparison with the normal distribution, as shown in Figure 15 [17], it can be seen that the CDF of the normal distribution at *x* = 1 is about 20% higher than that of the measured inter-null distribution, and its subsequent rise is relatively smoother. The normal distribution is clearly no better than the exponential distribution for fitting the measured inter-null distribution.

Regarding the comparison with the Poisson distribution, as shown in Figure 16 [17], it can be seen that in the beginning (*x* < 20), the CDF stays close to zero for a while and then its initial rise (after *x* = 20) is quite steep. When the interval is about 40, it rises to close to 1. The Poisson distribution is clearly no better than the exponential distribution in fitting the measured inter-null distribution. 

## 4. Further Evaluation of the Proposed NTP Improvement 

### 4.1. Adaptive Clock Adjustment and Clock Skew Compensation

Our proposed NTP improvement consists of two parts—an adaptive clock adjustment and clock skew compensation. In this section, we quantify the effectiveness of each part. The data set used in the evaluation is from 5400 NTP packet exchanges in a day, with 16 s intervals and an RTT threshold of 1.7 ms. Table 4 shows that the maximum offset value will far exceed the 1 ms objective if the adaptive clock adjustment is not adopted, and the average offset value will be higher than 0.2 ms. If the adaptive clock adjustment is adopted, the maximum offset value can be controlled within the 1 ms objective, with or without the clock skew compensation. However, if the clock skew compensation is applied on top of the adaptive clock adjustment, the average offset value can be improved from 0.109 ms to 0.043 ms. 

### 4.2. Synchronization between Clients

So far, we have been considering the clock synchronization between a client and a server. Now, we investigate the effectiveness of the proposed NTP improvement at synchronizing two clients. We consider two client machines, A and B, located in the same local area network as the regional time server. Clients A and B synchronize to the regional time server by themselves. By observing the difference between their recently derived offsets, we have some idea about whether and how the two client machines are synchronized [18].

Table 5, Table 6, Table 7 and Table 8 show that our NTP improvement can keep the synchronization accuracy between the clients within 0.1 ms in cases of 16 s, 32 s, and 64 s intervals. In the case of a 128 s interval, the maximum offset difference exceeds 0.2 ms. Its average offset difference is also larger than those of the three other cases. Nonetheless, all four cases can synchronize the two clients within the 1 ms objective. 

## 5. Conclusions

Because of its low cost and easy installation, NTP is widely used for time synchronization in the industrial Internet of Things (IIoT). The commonly used NTP mechanism does not take into account that the network packet travel time over a link is time-varying, which causes NTP to make incorrect synchronization decisions. Therefore, in this paper, we propose a simple, practical, and low-cost modification to NTP with clock skew compensation and adaptive clock adjustment by skipping the clock offset calculation when the NTP packet run trip time (RTT) exceeds a threshold, so that the NTP synchronization time can be controlled within 1 ms in the wired network environment.

First, we install the NTP mechanism and establish an experimental environment similar to a small mechanical factory (about 50 machine tools and a regional time server) and test this in a local manufactory network. The measurement results in a wired network environment show that the appropriate RTT threshold is 1.7 ms, and the NTP packet exchange interval can be 16, 32, or 64 s. Using one of the three intervals, the average offset can be controlled in the range of 0.008 to 0.02 ms, and the maximum offset is 0.131 ms. When the packet exchange interval is 128 s, the average offset rises to 0.037 ms, and the maximum offset reaches about 0.18 ms. Nonetheless, using the improved NTP proposed in this paper, in the environment of the wired network, the clock difference can be controlled within 1 ms, which is the required synchronization accuracy for the industrial Internet of Things.

Finally, we explain that the proposed adaptive clock adjustment is safe, and prove that the number of consecutive nulls is within a very safe range with a high probability, which will not cause the client clock to deviate from the server clock by more than 1 ms because of the accumulated clock skews.

## Figures and Tables

**Figure 1 sensors-22-05021-f001:**
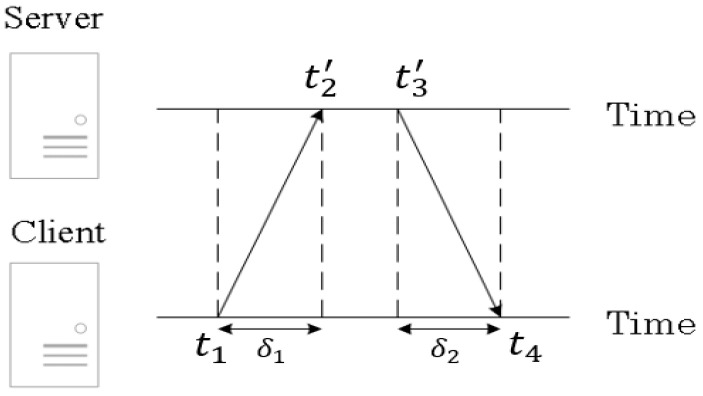
Packet exchange between the NTP client and NTP server.

**Figure 2 sensors-22-05021-f002:**
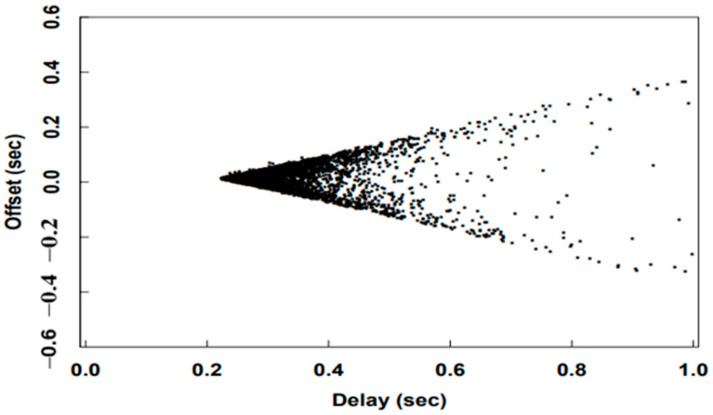
The relationship between clock offset and round-trip delay [7].

**Figure 3 sensors-22-05021-f003:**
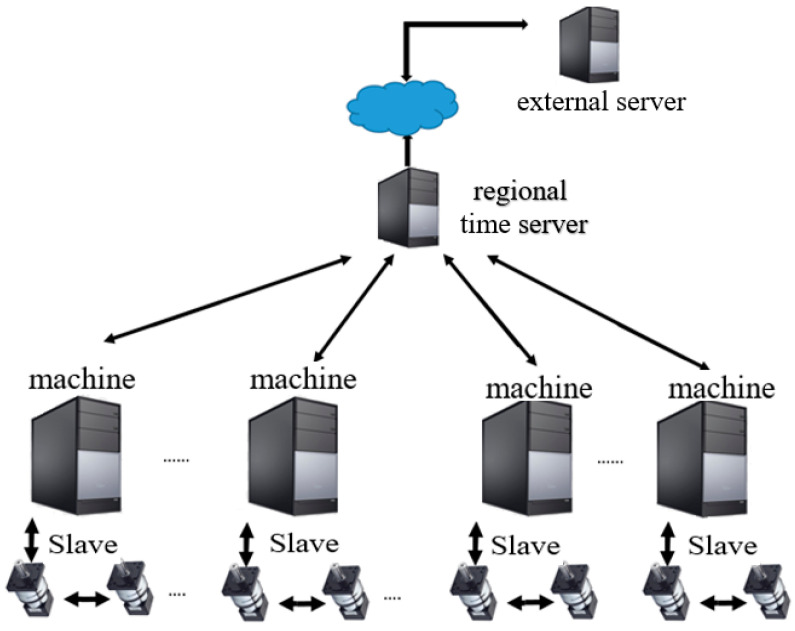
A reference architecture for NTP operation and measurement in a wired network environment.

**Figure 4 sensors-22-05021-f004:**
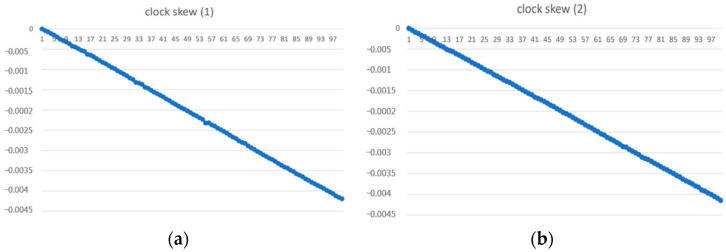
The results of the clock skew measurement. (**a**) Measurement results at 10:00. (**b**) Measurement results at 14:00. (**c**) Measurement results at 18:00. (**d**) Measurement results at 22:00.

**Figure 5 sensors-22-05021-f005:**
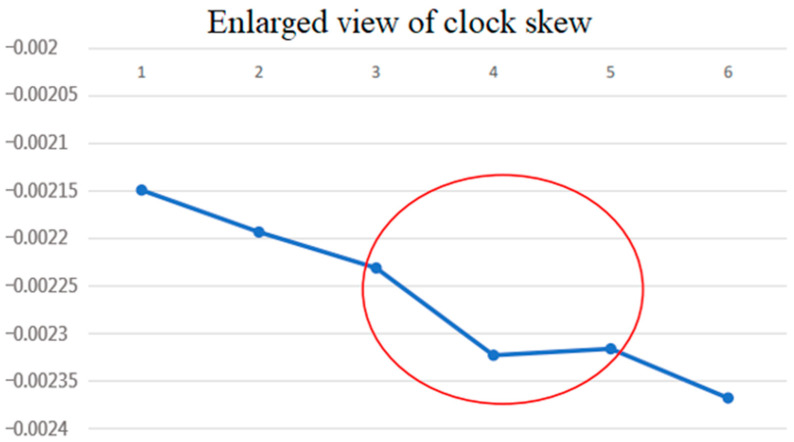
Enlarged view of the clock skew measurement (red circle-network delay).

**Figure 6 sensors-22-05021-f006:**
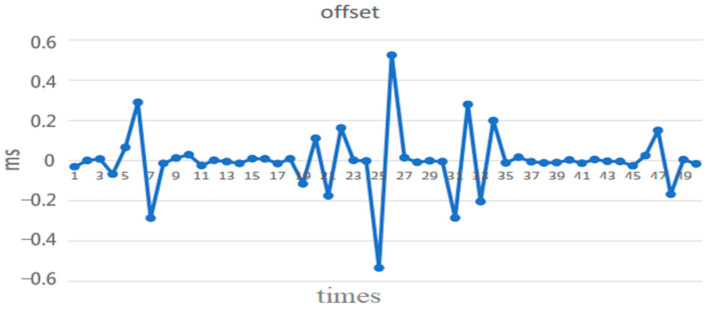
Offset measurement results from our NTP.

**Figure 7 sensors-22-05021-f007:**
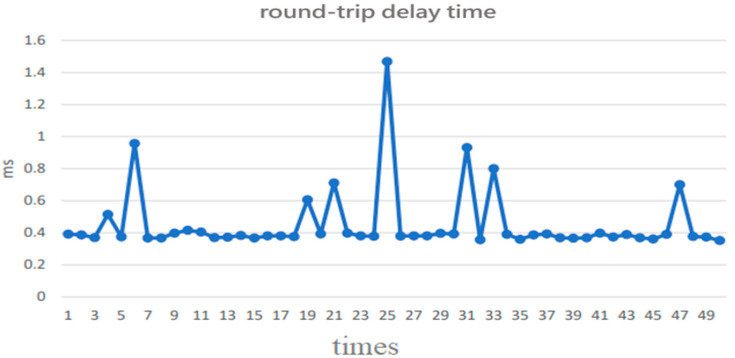
Round-trip time (RTT) measurement results from our NTP.

**Figure 8 sensors-22-05021-f008:**
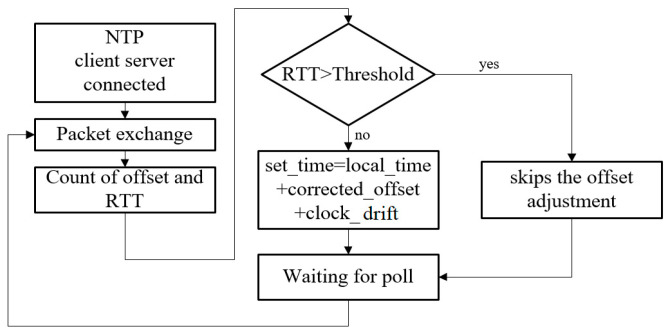
Flow chart of adaptive clock adjustment [14].

**Figure 9 sensors-22-05021-f009:**
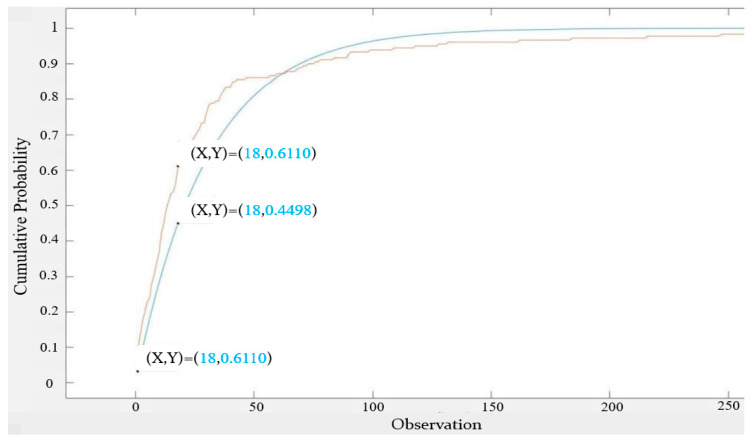
CDF of 16 s-based inter-null distribution vs. CDF of the exponential distribution with identical expected value (30.128).

**Figure 10 sensors-22-05021-f010:**
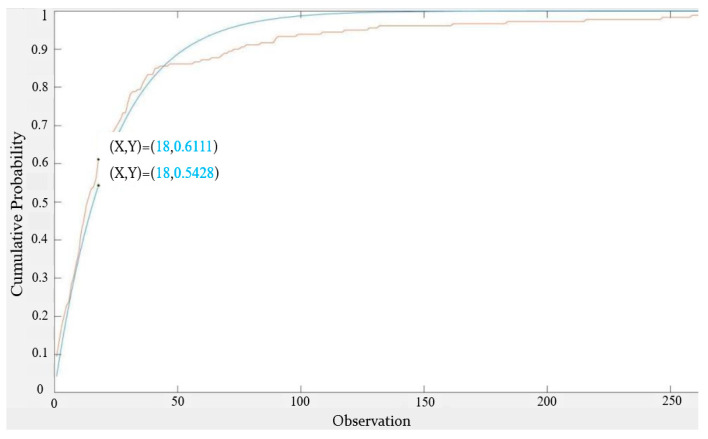
CDF of 16 s-based inter-null distribution vs. the CDF of the exponential distribution with an expected value = 23.

**Figure 11 sensors-22-05021-f011:**
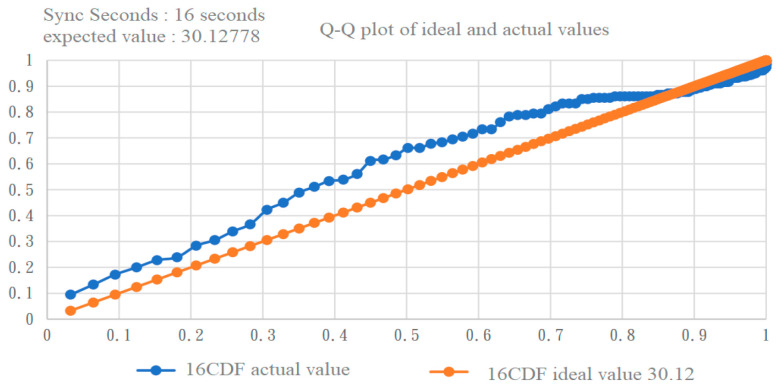
Q–Q plot of inter-null distribution vs. exponential distribution with an expected value of 30.128 [17].

**Figure 12 sensors-22-05021-f012:**
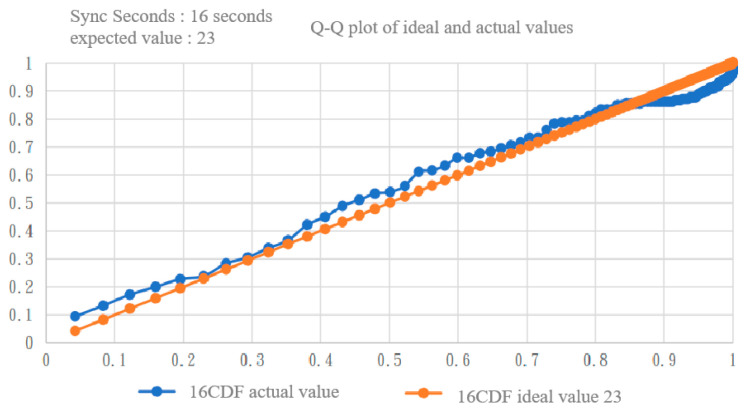
Q–Q plot of inter-null distribution vs. exponential distribution with an expected value of 23 [17].

**Figure 13 sensors-22-05021-f013:**
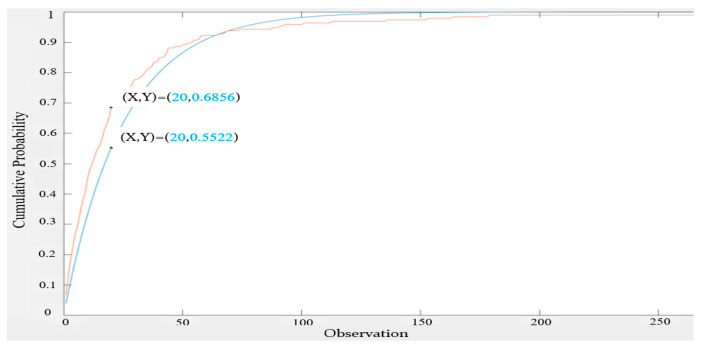
CDF of 32 s-based inter-null distribution vs. CDF of the exponential distribution with an identical expected value (24.897).

**Figure 14 sensors-22-05021-f014:**
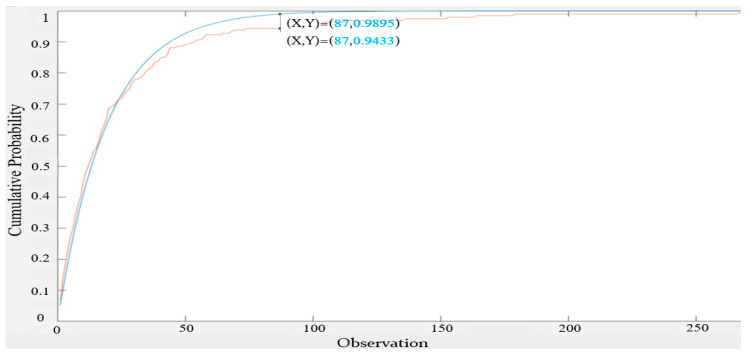
CDF of 32 s-based inter-null distribution vs. CDF of the exponential distribution with an expected value = 19.1.

**Figure 15 sensors-22-05021-f015:**
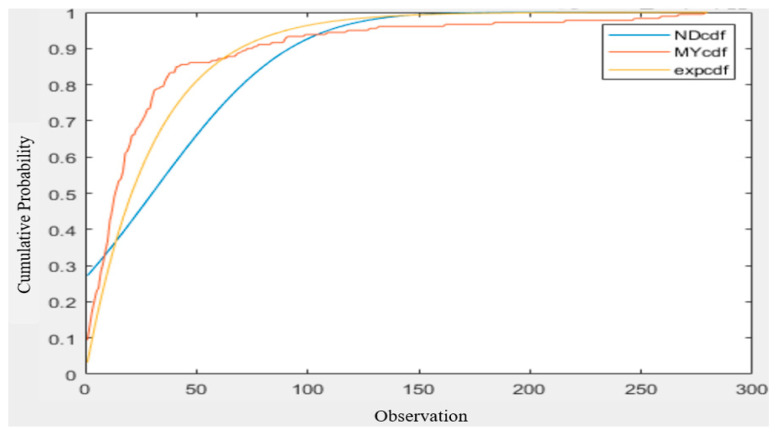
Comparison of normal distribution, exponential distribution, and measured inter-null distribution [17].

**Figure 16 sensors-22-05021-f016:**
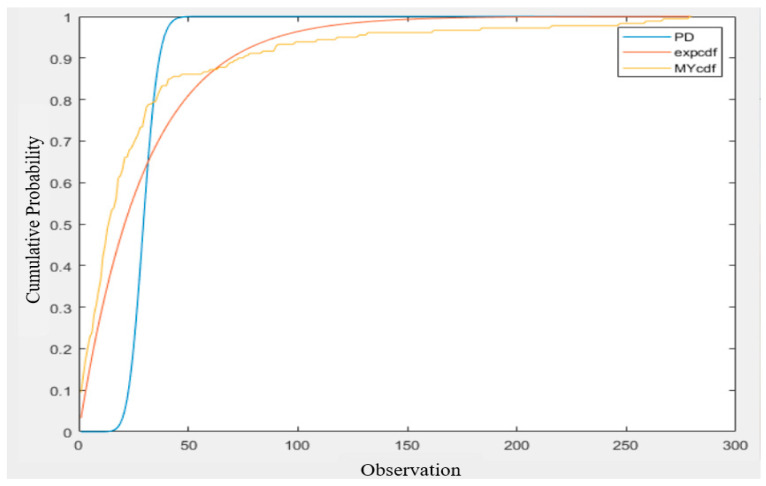
Comparison of the Poisson distribution, exponential distribution, and measured inter-null distribution [17].

**Table 1 sensors-22-05021-t001:** Comparison of offset and RTT with or without the removal of data sets when RTT exceeds the threshold.

255 Data in 1 h	A	B	C
offset_(*avg*)_ (ms)	0.489	0.206	0.046
offset_(*max*)_ (ms)	5.613	3.602	0.516
RTT_(*avg*)_ (ms)	0.949	0.604	0.411
RTT_(*max*)_ (ms)	11.587	7.335	1.428
Null percentage	-	-	7.5%

**Table 2 sensors-22-05021-t002:** Offset and null percentage at different null threshold values.

Null Threshold	offset_(*avg*)_ (ms)	offset_(*max*)_ (ms)	RTT_(*avg*)_ (ms)	RTT_(*max*)_ (ms)	Null Percentage
Non-Null	0.464	7.083	0.714	11.906	none
1.5 ms	0.045	−0.726	0.412	1.49	7.09%
1.6 ms	0.028	−0.77	0.398	1.594	4.38%
1.7 ms	0.023	0.656	0.387	1.683	3.34%
1.75 ms	0.035	−0.8	0.392	1.747	4.57%
1.8 ms	0.034	−0.73	0.389	1.8	2.50%
1.85 ms	0.037	−1.19	0.399	1.847	3.46%

**Table 3 sensors-22-05021-t003:** Offset, RTT, and null percentage statistics of a one-day measurement.

All Day Threshold 1.7 ms	16 s Sync	32 s Sync
offset_(*avg*)_ (ms)	0.023	0.054
offset_(*max*)_ (ms)	0.656	−0.716
RTT_(*avg*)_ (ms)	0.387	0.375
RTT_(*max*)_ (ms)	1.683	1.675
Null Percentage	3.34%	4.02%

**Table 4 sensors-22-05021-t004:** With or without adaptive clock adjustment and clock skew compensation.

Adaptive Clock Adj.	w/o		w/	
Clock Skew Comp.	w/o	w/	w/o	w/
offset_(*avg*)_ (ms)	0.224	0.261	0.109	0.043
offset_(*max*)_ (ms)	−5.64	11.577	0.759	−0.679
Null	w/o	w/o	5.3%	4.3%

**Table 5 sensors-22-05021-t005:** Sixteen second synchronization (based on server).

All Day 5400 Data	Client A	Client B	A-B
offset_(*max*)_ (ms)	−0.104	0.131	−0.12
offset_(*avg*)_ (ms)	0.019	0.02	0.023

**Table 6 sensors-22-05021-t006:** Thirty-two second synchronization (based on server).

All Day 2725 Data	Client A	Client B	A-B
offset_(*max*)_ (ms)	−0.066	0.077	−0.084
offset_(*avg*)_ (ms)	0.008	0.012	0.013

**Table 7 sensors-22-05021-t007:** Sixty-four second synchronization (based on server).

All Day 1370 Data	Client A	Client B	A-B
offset_(*max*)_ (ms)	0.079	0.062	−0.075
offset_(*avg*)_ (ms)	0.013	0.012	0.019

**Table 8 sensors-22-05021-t008:** One hundred and twenty-eight second synchronization (based on server).

All Day 680 Data	Client A	Client B	A-B
offset_(*max*)_ (ms)	0.128	0.18	0.214
offset_(*avg*)_ (ms)	0.03	0.037	0.058

## Data Availability

Not applicable.

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
