# Peer review of "An Improved Network Time Protocol for Industrial Internet of Thingsâ€"

_sensors, 2022, doi:10.3390/s22135021_

Round 1
Reviewer 1 Report
Abstract:
Failed to establish the real problem statement, method for proposed scheme and its outcomes. Furthermore, Lines 16-19 are grammatically incorrect and it is almost impossible to understand the what author intends to convey.
Introduction:
Lines 30,37
As the authors claimed that the Industrial Time synchronization requires (micro-seconds) and Industrial Internet of Things require (<1ms) delay, how can the proposed 1ms delay solution address this problem?
The rest of the introduction section failed to establish the problem statement and align the proposed method to it. In my suggestions, the authors should incorporate the following structure:
1. Define clear background of the topic
2. Define the problem statement
3. Clearly explain the current strategies to this problem (pros/cons)
4. Align the proposed solution to address these challenges
Furthermore, the equations can be moved to the methods section.
Proposed Method
It is very difficult to analyze the trend given in Figure 4. If the authors are recording the clock drift at various indexes, it could have been a lot simpler just to plot the variations using bars. Currently, the constant slope is very hard to understand.
Lines 118 : 122
The authors claimed that due to some network jitters the round trip time suffered that could result in synchronization errors. The authors further claimed that their proposed techniques nullifies any abnormal values, and thus ignores to synchronize at these indexes.
My question is that network delays and jitters are mostly uncertain and may arise due to a number of factors, including traffic pattern, interference, buffer overload, overhearing, re-transmission etc. How can the proposed scheme chooses to select the appropriate index over which they need to synchronize time? Furthermore, as the network jitter and delays are continuous, how the proposed scheme adaptively keeps track of these delays, and thus avoids synchronization errors?
In Figure 6 , offset measurements of NTP are presented that clearly show a significant change. How this change did not effect the drift as represented in Figure 7?
Lines 135-137
What do you mean by “these are more reasonable results”
The rest of the methods section is also confusing and failed to establish a clear direction of proposed work.
Results
The results and conclusion section is very confusing. There is no concrete pattern to analyze the results. Furthermore, the results are not compared against the most recent strategies. Finally, the conclusion failed to establish the viability of the proposed scheme.
Reviewer 2 Report
The paper utilizes an observation on the relation between clock offset and RTT and proposes an improved version of NTP. Overall, the technique is simple and practical and is shown to be effective. There are some issues need to be addressed before it is accepted for publication.
1. The theoretical analysis misuses the term "expectation."
2. There seems no evidences or arguments to support that the scheme can work in a more general environment.
3. It is unclear whether the clock skew is estimated on-the-fly (is it necessary?).
4. The placement of references should be edited.
Other minor comments are included in the attached file.

Reviewer 3 Report
The paper proposes a modification to the synchronization protocol NTP. The results show an adequate performance and improvement related to the original NTP mechanism.
However, the contribution of the paper and the proposed modification are not clearly stated and described. The authors should explain in better detail what the modification is and how it works.
Round 2
Reviewer 1 Report
Thanks for revising and improve the quality of the article. However, some points have not been addressed clearly:
-What the authors means of 'We will use these data to find a fitted probability distribution function 301 and adopt a reasonable mathematical model to prove our claim'
- Explain in details about the 'server'
- Regarding my previous question :
'network delays and jitters are mostly uncertain and may arise due to a number of factors, including traffic pattern, interference, buffer overload, overhearing, re-transmission etc. How can the proposed scheme choose to select the appropriate index over which they need to synchronize time?'
Please explain/describe in the article.
Reviewer 3 Report
The authors have considered my previous concerns and they have corrected the manuscript accordingly. As such, I believe that it has enough merits to be published in its current form.
Author Response
Thank you for the comments.